# Assessment of Food Safety Knowledge and Behaviors of Cancer Patients Receiving Treatment

**DOI:** 10.3390/nu11081897

**Published:** 2019-08-14

**Authors:** Holly Paden, Irene Hatsu, Kathleen Kane, Maryam Lustberg, Cassandra Grenade, Aashish Bhatt, Dayssy Diaz Pardo, Anna Beery, Sanja Ilic

**Affiliations:** 1Department of Human Sciences, Ohio State University, Columbus, OH 43210, USA; 2Department of Internal Medicine, College of Medicine, Ohio State University, Columbus, OH 43210, USA; 3Department of Radiation Oncology, College of Medicine, Ohio State University, Columbus, OH 43210, USA

**Keywords:** cancer, food safety, food insecurity, food safety knowledge, foodborne disease

## Abstract

Cancer patients receiving treatment are at a higher risk for the acquisition of foodborne illness than the general population. Despite this, few studies have assessed the food safety behaviors, attitudes, risk perceptions, and food acquisition behaviors of this population. Further, no studies have, yet, quantified the food safety knowledge of these patients. This study aims to fill these gaps in the literature by administering a thorough questionnaire to cancer patients seeking treatment in three hospitals in a Midwest, metropolitan area. Demographic, treatment, food security, and food safety knowledge, behaviors, attitudes, risk perceptions, and acquisition information was assessed for 288 patients. Specific unsafe attitudes, behaviors, and acquisition practices were identified. Most notable is that 49.4% (*n* = 139) of participants were not aware that they were at increased risk of foodborne infection, due to their disease and treatment. Additionally, though patients exhibited a general understanding of food safety, the participant average for correctly answering the food safety questions was 74.77% ± 12.24%. The section concerning food storage showed lowest participant knowledge, with an average score of 69.53% ± 17.47%. Finally, patients reporting low food security also reported a higher incidence of unsafe food acquisition practices (*P* < 0.05). These findings will help healthcare providers to better educate patients in the food safety practices necessary to decrease risk of foodborne infection, and to provide targeted food safety education to low-food-security patients.

## 1. Introduction

With an estimated 48 million people sick, 128,000 hospitalized, and 3000 dead from foodborne disease every year in the US [1], implementation of proper food safety practices among the general consumer population is of critical importance for public health. Foodborne diseases occur due to consumption of foods contaminated with foodborne viruses, bacteria, or micro parasites. The most common foodborne pathogens include norovirus, nontyphoidal *Salmonella* spp., *Campylobacter* spp., and others. *Listeria monocytogenes* causes some of the most severe infections, with up to 30% death in immunocompromised patients [1].

Cancer patients experience increased susceptibility to foodborne illness, compared to people under the age of 65 with no preexisting conditions [2]. This is dependent on the patient’s diagnosis and the pathogen to which they are exposed. A patient diagnosed with gynecological cancer is at 66-times greater risk for infection by *Listeria monocytogenes* than someone from the general population, while a patient diagnosed with a blood cancer is 1364 times more susceptible to infection [3]. Cancer patients are very susceptible to *Saccharomyces cerevisiae* [4] and *Toxoplasma gondii* [5], due to immunosuppression, and experience highly aggressive infection by *Escherichia coli* when exposed to the bacterium [6]. The incidence of cancer remains high, with an estimated 1,762,450 new cancer diagnoses and 606,880 resulting deaths in 2019 alone [7].

Patients experiencing any stage of cancer and those receiving any form of treatment (e.g., chemotherapy, radiation therapy, etc.) are at increased risk of contracting foodborne infections due to treatments affecting and depressing immune system mechanisms [2]. Additionally, patients may experience post-treatment immune suppression known as “neutropenia” [8]. This state is defined as having ≤1000 neutrophil cells per µL of blood [9] and occurs due to the mechanisms of radiation therapy and chemotherapy [10]. Because these two therapies target rapidly dividing cells, cells native to the host immune system are also destroyed [10], resulting in a severely immunocompromised state following each administration of therapy [8]. Therefore, for this population, appropriate food safety practices are vital.

Certain behaviors and food choices have been shown to decrease the risk of contracting foodborne illness [11,12]. To minimize the risk, it is recommended that consumers wash their hands and clean kitchen surfaces, prevent cross-contamination from raw foods, cook to safe temperatures, and refrigerate foods promptly and properly [13]. On the other hand, food choices that increase risk include raw or undercooked fish or shellfish (such as sushi or ceviche), partially cooked seafood, unpasteurized milk, raw or undercooked eggs, raw sprouts, unwashed fresh vegetables (particularly leafy greens), and meat products (e.g., hot dogs and deli meats) that have not been reheated [10]. Finally, recommendations for safe food acquisition practices include avoiding foods that are past the sell-by date, damaged canned goods, wilted or damaged produce, and avoiding foods that are displayed in unsanitary locations [14].

The studies assessing the food safety knowledge and practices of cancer patients in treatment are scarce [10,15,16], and show general low food safety knowledge [15] among this population. The available evidence demonstrates that cancer patients may not perform appropriate food safety behaviors [10]. Even when the patients were aware of the increased food safety risks due to their condition [17], they did not link their awareness of increased susceptibility for infection with their routine food handling practices [17]. Studies in general consumer behavior, that include an aging population demographically similar to cancer patients (>65 years of age), show the increased risk for errors in food safety behaviors [16]. 

The presence of financial strain can impact the amount of money allotted for food, with implications to food security and food safety status. It has been found that food insecurity leads to increased risk of exposure to foodborne pathogens. For instance, food-related coping mechanisms utilized by people with limited resources, such as removing spoiled parts of produce, removing insects from grains, or consuming slimy meat products, have been demonstrated to increase food safety risks [14]. Eating other people’s leftovers and consuming roadkill are some examples of risky food acquisition practices identified in food-insecure populations, in which the consumer cannot guarantee the safety or cleanliness of the foods [14]. Financial burden of cancer treatment may lead to increased risk of food insecurity, as certain factors including low socioeconomic status, low income, and insufficient or absent health insurance have been shown to be associated with poverty, and thus with food insecurity [18,19]. Food insecurity may become a hurdle for a patient receiving cancer treatment, increasing their risk of contracting foodborne infections [14,20].

By raising awareness and informing cancer patients of the increased risk of foodborne illness, it is possible to decrease risky behaviors and practices [17]. However, patients often do not receive the food safety information. Even when education is received, it varies depending on the treatment facility [15]. In addition, the patients may not understand how to put the given recommendations into practice, despite expressing willingness to follow the guidelines [17]. A more thorough analysis of food safety behaviors, attitudes, and beliefs is needed to better assess factors that may contribute to patient food safety knowledge, in order to develop a more appropriate approach to cancer patients’ food safety education.

The objective of this study was to determine current food safety knowledge, behaviors, and attitudes among cancer patients seeking treatment, and to assess the effect of sociodemographic factors, including food insecurity, on food safety risks. The findings of this study will allow for clear recommendations and the development of more effective strategies to reduce foodborne infections among cancer patients receiving treatment and ultimately improve health outcomes, wellbeing, and quality of life in this population. 

## 2. Materials and Methods

### 2.1. Study Participants

This is an observational, cross-sectional study, performed in accordance with the Declaration of Helsinki. It was approved by the Ohio State University Institutional Review Board with the project identification code 2016C0013. Cancer patients receiving treatment were selected by consecutive convenience sample. Patients, diagnosed with any stage (I–IV) or variant of cancer and receiving any treatment, were recruited from three cancer clinics in Columbus, Ohio, USA between June 2016 and August 2017. These clinics were the OSU James Cancer Hospital and Solove Research Institute, OSU Hospital East, and the Stefanie Spielman Comprehensive Breast Center. Methods for recruitment included referrals, word of mouth, and posting on an online study recruitment website: studysearch.osumc.edu. In order to be eligible to participate in the study, patients had to satisfy the following requirements: be at least 18 years of age, clinically diagnosed with cancer, currently receiving some form of cancer treatment (e.g., surgery, radiation, chemotherapy, hormone therapy, combination therapy, etc.), and willing and able to give written, informed consent. 

Patients were approached at cancer clinics by trained research assistants and were given a thorough description of the research protocol and procedures. Consenting patients were given self-administered questionnaires, with assistance provided by the research assistants when needed. Upon completion of the study survey, participants received a $15 gift card to a local grocery store chain as an incentive for their participation.

### 2.2. Measures

#### 2.2.1. Socio-Demographic Characteristics, Food Insecurity, Disease and Treatment

A comprehensive questionnaire was designed to assess socio-demographic characteristics of the participating cancer patients. Complete questionnaire is in Appendix A. The survey detailed participants’ gender, age, ethnicity, marital status, education, employment status, household income, size, number of children, health insurance status, and food assistance. Participants smoking, alcohol and drug use status were surveyed. In order to assess the presence and the extent of financial burden due to cancer treatment, timeliness of bill payments and the necessity to borrow money were included in the questionnaire.

A short construct was used to assess patient disease characteristics, including type and stage of cancer, time since diagnosis, and the treatment received [21]. The food security status among cancer patients was determined using US Adult Food Security Module of the USDA, ERS screening tool [22]. The module was adapted to the survey context and integrated into the questionnaire. 

#### 2.2.2. Food Safety Assessment

Food safety risk perception, attitudes towards food safety, and food safety behaviors were assessed using five-point Likert scales. A four-item scale assessed patient risk perception, with response options ranging from “strongly disagree” to “strongly agree”. Attitudes were assessed on a modified 14-item evaluated scale, and a 15-item scale was used to measure patient behaviors. Both of these sections offered response options ranging from “never” to “always” [17].

The consumption of high-risk foods was assessed using a modified construct containing 13 items and a binary response scale [10]. Common high-risk raw and ready-to-eat (RTE) foods (such as rare hamburgers, runny egg yolks, soft cheeses, etc.) were included in the questionnaire. High-risk acquisition behaviors were assessed using a 25-item survey on a five-point Likert scale (1 = never, 5 = always) developed by Anater et al. [23], which included questions concerning expiration dates, damaged packaging, and consumption of non-food items.

Food safety knowledge questions (n = 45) were organized in statements grouped in five sections: general food safety, cross-contamination, food preparation, food storage, and clean up. The participants could agree or disagree with each given statement. The responses were coded on a binary scale as correct/incorrect and scored as described in the statistical analysis section. General food safety knowledge was assessed using the construct consisting of 11 statements, designed to probe participants’ basic knowledge of foodborne human pathogens. A total of eight questions tested the participant knowledge of cross-contamination and separation, while food preparation, food storage, and cleaning and hygiene were tested using constructs that included eight, eight, and 10 questions, respectively [10,11,17].

### 2.3. Statistical Analysis

Data were analyzed using IBM SPSS Statistics version 25. All responses were tabulated, and the data were analyzed descriptively using frequencies. The sections for food security status were scored as previously described [22]. Using a validated scoring system [24], patients were either determined to be (i) food secure, (ii) marginally food secure, (iii) have low food security, or (iv) have very low food security. Frequencies were generated for each Likert scale item [25]. The responses for food preferences were also presented as frequencies. A food acquisition penalty score was calculated wherein any participant response other than “never” resulted in a coding of 1 point. A higher overall food acquisition score meant a more frequent demonstration of unsafe food acquisition practices. Food safety knowledge responses were scored by assigning one point for each correct response, and 0 points for each incorrect or a lack of response. The scores were expressed as sums per each food safety knowledge section, and cumulatively, as a total food safety knowledge score, with 45 points being the maximum and 0 points the minimum score. Dichotomous data and Likert data was summarized and analyzed separately. Food acquisition scores and overall percent for food safety knowledge were compared to demographic characteristics, food security score, and disease characteristics using one-way ANOVA.

## 3. Results

### 3.1. Demographics, Diseases Characteristics, and Food Security

Of the 288 cancer patients who participated in the study, most were aged 50 years or older (77.1%, *n* = 222) and female (66.9%, *n* = 192) (Table 1). About one-third of patients had a college degree or more (36.8%, *n* = 106). The majority reported a household monthly income of less than $4000 a month (64.2%, *n* = 179), which is below the median Ohio monthly income of $6315 [26]. A total of 12.9% (*n* = 37) of the recruited patients participated in some form of food assistance program and 9.8% (*n* = 28) received food from a food bank, food pantry, or soup kitchen.

Nearly half the patients (48.7%, *n* = 140) were current or former smokers, 3.5% (*n* = 10) used recreational drugs, and 40.6% (*n* = 117) drank alcohol.

The majority of patients had been diagnosed with cancer within the past 6 months (64.1%, *n* = 184), and more than one-quarter were in the fourth stage of the disease (29.7%, *n* = 84). The majority of patients were receiving a combination of two or more therapies (63.9%, *n* = 184), and 29.9% (*n* = 86) were receiving chemotherapy. Almost half (45.6%) of patients also received an oral medication as part of the treatment (*n* = 131). More than a quarter of patients had been diagnosed with breast cancer (38.6%, *n* = 105) (Table 2).

While the majority of patients were food-secure (80.9%, *n* = 233), 9.4% (*n* = 27) reported marginal or low to very low (9.7%, *n* = 28) food security. Complete patient response frequencies are shown in Appendix A.

### 3.2. Risk Perceptions, Attitudes, and Behaviors

In assessing risk perceptions, over half of the surveyed population thought that food contamination with foodborne pathogens was a serious problem (70.2%, *n* = 202) (Figure 1a). However, approximately half were not aware that they were at increased risk of contracting a foodborne infection due to their condition and treatment (49.4%, *n* = 139). Nearly all participants thought they knew how to keep food safe at home (91.3%, *n* = 263) or when eating out (83.7%, *n* = 241). Overall, patients had the most positive attitudes towards washing cutting boards, knives, and countertops after cutting raw meat with 87.4% (*n* = 249), stating that they liked to perform this practice.

Also, the majority reported the preference to store eggs in the refrigerator (76.4%, *n* = 217) (Figure 1b). Less positive attitudes were observed for certain vital food safety aspects. For instance, less than half reported concern over thawing perishables correctly (48.8%, *n* = 140). Similarly, only 47.0%, (*n* = 135) were interested in using a meat thermometer. The most negative attitudes expressed were towards drinking pasteurized apple juice or cider, where 56.7% (*n* = 160) did not feel that this was an important food safety practice. Only 43.4% (*n* = 123) of the surveyed cancer patients were worried that they would get sick if they ate raw hot dogs (Figure 1b).

Age (*P* < 0.05), income (*P* < 0.05), receiving assistance from federal food programs (*P*< 0.001), and food security status (*P* < 0.05) all were identified as factors with significantly different means between groups when considered using the food acquisition penalty score at α = 0.05.

In considering food safety behaviors, the majority reported washing their hands before preparing foods (95.1%, *n* = 269) (Figure 1c). Patients also consistently reported washing raw vegetables (90.2%, *n* = 257). They washed their hands (95.8%, *n* = 272) and plates (97.2%, *n* = 276) after handling raw meat, chicken or seafood, as well as countertops after food preparation (90.8%, *n* = 259). Most of the participants typically did not leave cooked foods out overnight on the stovetop (89.8%, *n* = 255), and did not leave eggs out at room temperature (91.2%, *n* = 259). Some participants reported thawing meat and poultry on the counter top (39.8%, *n* = 113). The majority of participants did not typically prepare or serve food for others when suffering from diarrhea (80.1%, *n* = 225). However, participants rarely tended to use a thermometer to monitor their refrigerator temperature (32.9%, *n* = 93), or to determine if chicken breasts had been sufficiently cooked (34.3%, *n* = 97). Cancer patients almost never used a thermometer while reheating leftovers (14.5%, *n* = 41).

### 3.3. Food Preferences and Food Acquisition Behaviors

More than half of participants consumed eggs with runny yolks (55.6%, *n* = 159), restaurant salad bar items (69.1%, *n* = 197), and cold deli meats (88.4%, *n* = 252). Participants consumed raw homemade cookie dough in 32.5% (*n* = 93) of cases, and soft cheeses in 38.6%, (*n* = 110) of cases. Less than a quarter participants indicated that they consumed raw sprouts (22.4%, *n* = 64), cold hot dogs (15.2%, *n* = 43), smoked fish served cold (13.7%, *n* = 39), sushi (16.2%, *n* = 46), raw oysters (12.9%, *n* = 37), rare hamburgers (8.7%, *n* = 25), raw fish (9.8%, *n* = 28), or ceviche (7.7%, *n* = 22).

Many surveyed patients participated in risky food acquisition behaviors, specifically removing spoiled parts from fruits and vegetables before consumption (46.3%, *n* = 131) and cooking food with other people (84.9%, *n* = 242). Although rarely observed, the following food acquisition behaviors were identified: seeking out (2.1% *n* = 6) or eating (1.2%, *n* = 3) roadkill, purchasing expired foods (10.2%, *n* = 29), eating expired food (21.5%, n = 61), removing slime from lunch meat (13.1%, *n* = 37), removing mold from cheese (36.2%, *n* = 103), removing mold from grains (10.6%, *n* = 30), removing insects from grains (15.6%, *n* = 44), eating non-food items (5.7%, *n* = 16), and eating pet food (1.1%, *n* = 3).

### 3.4. Food Safety Knowledge

Food safety knowledge scores were overall low among cancer patients, with an average score of 74.77 ± 12.24%, (average ± standard deviation) (Table 3). Food storage safety was the most poorly understood section (69.53 ± 17.47%). In the general food safety knowledge section, the most common misconceptions included thinking that pesticide residues are the most serious food safety problem (66.3%, *n* = 187) and that unsafe foods can be identified by the way they look or smell (55.3%, *n* = 157). Concerning cross-contamination, most participants did not know that wiping off a cutting board with a wet dishcloth or sponge, after cutting raw meat, is not sufficient to prevent cross-contamination (60.1%, *n* = 170). In food preparation safety, over half of participants thought that one can determine that a hamburger has been cooked completely based on its color (53.5%, *n* = 152). A majority of participants believed that it is safe to leave hot foods to cool to room temperature on the counter, before refrigerating them (67.0%, *n* = 189), and that green bean casserole can be safely consumed if reheated properly after being left out overnight (92.5%, *n* = 260). Lastly, most participants believed that soap and water were sufficient to sanitize countertops (79.9%, *n* = 226).

Using one-way ANOVA, income (*P* < 0.005), enrollment in a federal food assistance program (*P*< 0.001), enrollment in a private food assistance program (*P* < 0.001), smoking frequency (*P* < 0.05), and food insecurity (*P* < 0.001) were identified as significant factors for overall percent food safety knowledge (Table 4).

## 4. Discussion

This study assessed and quantified food safety knowledge among cancer patients who are currently in treatment. When patients begin treatment, the weakening of their immune system increases their risk of contracting foodborne illness to up to >1000 times the risk of the general population [3]. Having an understanding of patients’ knowledge, attitudes, behaviors, and risk perceptions is vital to the development of appropriate food safety education to prevent the adverse effects that foodborne diseases could impose on this population. While previous studies have considered some of these aspects, this is the first time they have all been investigated in conjunction with food acquisition practices and food insecurity among cancer patients.

We found that only one-half of studied cancer patients were aware of the increased risk of contracting foodborne disease due to their condition and treatment, despite the fact that all of them received at least one dietary assessment, and follow-up nutritional therapy given by a dietitian. Similar to our findings, patients surveyed in an earlier study did not believe that they were more susceptible to infection than other, healthier adults [10]. In addition, it has been previously reported that cancer patients receiving treatment were not given any food safety advice by healthcare professionals, unless they were actively experiencing neutropenia [17], and felt they needed to receive food safety information much earlier, preferably shortly after diagnosis and as soon as the first scheduled oncology appointment [17]. All this indicates that the food safety information that cancer patients currently receive is, in general, insufficient to ensure their awareness of food safety risks. Patients’ responses to our study emphasize the need for improved education at the beginning of cancer treatment, in order to better mitigate the risks of foodborne illness.

Cancer patients’ inadequate prioritization of food safety during cancer treatment, in this study, may be due to their low perception of risk or their low level of interest in food safety activities. The Health Belief Model posits that people will take action to protect themselves from health-based dangers if they are under the assumption that their actions will have an effect and if they believe that they can properly execute the actions [27]. Most patients, in this study, believed that they knew how to keep food safe both in the home and outside of the home. However, based on the results above, patients exhibited only a moderate understanding of overall food safety, with the lowest understanding of safe food storage practices. This demonstrates that patients are overestimating their food safety understanding and may be putting themselves at risk by not taking the appropriate steps to keep themselves safe while eating at home or eating out.

We found that income, enrollment in food assistance programs, and food security were factors associated with both food acquisition practices and food safety knowledge. Enrollment in federal and private food assistance programs, as well as general food insecurity, were previously reported to contribute to risky food management practices [14]. It should also be noted that a low income has been previously associated with an increase in the incidence of foodborne disease [28]. While some of this has been attributed to high-risk foods consumed by low-income populations, these groups show gaps in food safety knowledge that puts them at higher risk for acquisition of foodborne illness [28]. Similar to our study, two previous papers identified problems with the food safety knowledge in low-income populations pertaining to thermometer use and temperature [29,30]. Consumption of undercooked eggs has been identified in our study and in previous studies of low-income adults [30]. Lack of understanding pertaining to how to properly sanitize kitchen implements was found in our study and shown to be common amongst participants in the United States Women, Infant, and Children (WIC) food assistance program, in a previous study [29]. 

Lack of sufficient food safety knowledge and a lack of adherence to proper hygiene practices, which would ensure food safety in surveyed cancer patients, was similar to the general population [16]. This knowledge combined with these practices is important in cancer patients because improper food management may have more severe consequences to health outcomes in this population. Patient responses in our study clearly demonstrate low risk perception, as well as low motivation to implement certain food safety practices. In line with this, reported behaviors are often not appropriate to achieve food safety. For instance, the attitudes around the use of food thermometers ranged from ambivalent to negative. Accordingly, patients reported infrequent use of food thermometers in their homes. This specific behavior was shown to be problematic in previous reports [17], highlighting the need to design an intervention to effectively address the use of thermometers.

Further, even when positive attitudes were present concerning certain food safety practices, gaps in understanding were identified that would place cancer patients at increased risk of foodborne illness. Participants in our study had positive attitudes about storing eggs in the refrigerator, with the majority expressing that this was necessary (76.4%), which was reflected in the corresponding behavior of 91.2% of patients storing raw eggs in the refrigerator. However, less than half of the patients thought that it was important to thoroughly cook egg yolk, and over one half (55.6%) stated that they consumed eggs with runny yolks. Because eggs with runny yolks are undercooked and may still contain bacteria that can be harmful to immunocompromised individuals [10], future educational materials targeting cancer patients must include this specific information.

Nearly all patients (97.9%) in this study, reported consuming high-risk foods. The most commonly consumed high-risk foods were items from salad bars, with 69.1% of the surveyed patients saying they ate these foods. This can present a challenge, as many foods that are perceived as healthy may also be at high-risk for contamination with foodborne pathogens (e.g., fresh produce, sprouts, raw fish, fresh squeezed juices, etc.) [10]. Furthermore, it has been a common practice that cancer patients follow a neutropenic diet during their treatment to decrease the risk of foodborne illness [31]. However, recent evidence suggests that following food safety recommendations is more beneficial to cancer outcomes than the consumption of neutropenic diets [32], due to its better nutrition outcomes and consumption of bioactive compounds found in such foods [33]. Therefore, to increase knowledge of proper food safety handling techniques and change food safety behaviors would be effective not only in preventing foodborne infections but also improving treatment outcomes in this population.

Though infrequent, some patients reported extremely high-risk food acquisition behaviors related to coping strategies in food-insecure populations. Of the participants in this study, 19.1% reported marginal to very low food security. While this is higher than the state of Ohio food insecurity rate of 14.5% [34], there is no previously reported information on food insecurity rates in cancer patients in Ohio. In a study performed in Kentucky, cancer patients reported a food insecurity rate of 17.4% [35], compared to 14.9% in the general population of Kentucky [34]. This implies that cancer patients may suffer from a higher rate of food insecurity than the general population, emphasizing the importance of providing food-insecure patients with food safety information specific to the previously identified risky food acquisition behaviors. 

One limitation of this study is the skew of the sample presented due to convenience sampling in cancer centers. Most participants of this study reported a lower monthly household income than the median monthly household income of Ohio (<$4000 compared to $6315) [26]. Since one of the sampling locations was a hospital specializing in breast cancer treatment, a larger proportion of the patients than is representative of the general population of Ohio were women (66.9% compared to 51.4%) [36]. However, the sampled patients in this study closely reflect both the race and education composition of Ohio [26,37]. Additionally, our participants were nearly equal in the proportion of people in Ohio receiving Supplemental Nutrition Assistance Program (SNAP)/food stamps (sample = 11.2%, Ohio = 11.6%) [38], and the median age range of participants closely reflected the overall median age range of cancer patients at diagnosis [39]. Due to this close adherence to the demographics of the general population, the results of this study may be able to be generalizable to the Ohio cancer population. However, generalization of the results to areas outside of Ohio, or the Midwest, may not be applicable. Future studies should include questions concerning the previously identified behaviors in cancer patients and in other high-risk populations, as well as the information about the caregivers’ food safety knowledge and practices, in order to continue to expand the body of literature and improve educational materials targeted toward these people. 

## 5. Conclusions

The impact of this study is the identification of risky food acquisition and food safety behaviors in cancer patients, as well as the quantification of patient knowledge pertaining to food safety. Due to their susceptibility to foodborne illness, education of cancer patients is vital to decreasing risk and improving the outcome of treatment. Food safety education is particularly important in the first six months after diagnosis, as treatment begins. Establishing good food safety habits and acquisition practices in the early stages of treatment will help patients throughout the course of the disease. The need for proper food safety behaviors only increases in importance as patients progress to the higher stages of cancer, concurrently increasing their susceptibility to illness. Patients who are food insecure are at an even higher risk of infection, due to the performance of dangerous food acquisition behaviors. Though the behaviors identified in this study were not predominant within the cancer patient population, this data clearly show association between food insecurity and the identified acquisition practices.

## Figures and Tables

**Figure 1 nutrients-11-01897-f001:**
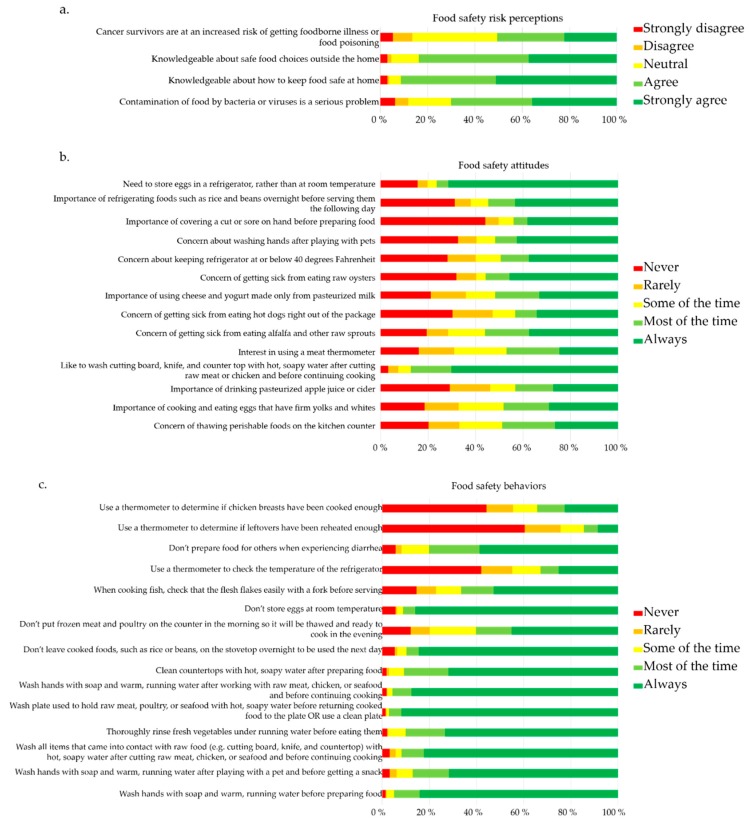
Percentile divisions of participant responses to (**a**) food safety risk perceptions, (**b**) food safety attitudes, and (**c**) food safety behaviors in the given survey.

**Table 1 nutrients-11-01897-t001:** Demographic Factors of Surveyed Patients (*N* = 288).

Demographic Characteristics	Categories	*n* ^a^	%
Gender	Female	192	66.90
	Male	95	33.10
Age	18–29	7	2.45
	30–39	19	6.64
	40–49	38	13.29
	50–59	97	33.92
	60–69	84	29.37
	>70	41	14.34
Race	Asian	2	0.70
	Black/African	27	9.41
	Hispanic	2	0.70
	Native American	3	1.05
	White	250	87.11
	Other	3	1.05
Marital status	Married	175	60.76
	Single	31	10.76
	Divorced/Widowed	77	26.74
	Other	5	1.74
Highest level of education	<H.S. ^b^	81	28.13
	H.S./GED ^c^	38	13.19
	1–2 college	63	21.88
	≥college	106	36.81
Employment status	40+ h/wk^d^	96	33.33
	≤40 h/wk^d^	40	13.89
	Home	19	6.60
	Retired	92	31.94
	None	41	14.24
What is your household’s monthly income? ($)	<$1000	25	8.96
	<$2000	53	19.00
	<$3000	56	20.07
	<$4000	45	16.13
	≥$4000	100	35.84
How many people live in your household?	1	55	19.1
	2	131	45.49
	3	44	15.28
	≥4	31	10.76
	≥5	27	9.38
How many children (<18 yrs.) live in your household?	0	209	72.60
	1	34	11.81
	2	26	9.03
	3	14	4.86
	≥4	5	1.74
What is your health insurance status?	None	4	1.39
	Private	143	49.83
	Public	39	13.59
	Medicare	101	35.19
Have you had to borrow money to pay for healthcare?	Yes	26	9.09
	No	260	90.91
Have you had to pay your bills late due to medical expenses?	Yes	69	24.13
	No	217	75.87
Which food assistance programs do you participate in?	SNAP ^e^/food stamps	32	11.19
	WIC ^f^	0	0
	Other	5	1.75
	None	249	87.06
Receive foods from a food bank, a food pantry, or a soup kitchen	Yes	28	9.79
	No	258	90.21
What is the best description for where you live?	Urban	55	19.57
	Rural	102	36.3
	Suburban	124	44.13
What is your smoking status?	Current	33	11.46
	Former	107	37.15
	Never	148	51.39
If current smoker, how often do you smoke?	0	251	87.46
	Daily	30	10.45
	4–6d/w ^f^	3	1.05
	2–3d/w ^f^	2	0.7
	1d/w ^f^	1	0.35
Do you use any recreational drugs?	Yes	10	3.47
	No	278	96.53
Do you drink alcohol?	Yes	117	40.63
	No	171	59.38
If yes, how often?	0	171	60
	Daily	19	6.67
	4–6d/w ^g^	16	5.61
	2–3d/w ^g^	29	10.18
	1d/w ^g^	50	17.54

^a^ Due to missing data for some factors, not all categories add up to *N* = 288, *n*: is the sample size for each factor. ^b^ H.S. is the abbreviation for “High School”. ^c^ General Educational Development (GED) is a collection of four subject tests that, when passed, yield a certification equivalent to a high school degree, in the United States and Canada. ^d^ Hours per week. ^e^ Supplemental Nutrition Assistance Program (SNAP) provides a stipend for food assistance to those who qualify. ^f^ Women, Infants, and Children (WIC) is a food assistance program specifically for pregnant, postpartum, and breastfeeding women, as well as infants and children under the age of five years old. ^g^ Days per week.

**Table 2 nutrients-11-01897-t002:** Cancer Diagnosis of Surveyed Patients.

Cancer type	*n* ^a^	%
Breast	105	38.6
Prostate	43	15.81
Ovarian	14	5.15
Cervical	15	5.51
Colon/Rectum	31	11.4
Lung	11	4.04
Other	53	19.49

^a^ Because of missing data for some demographic factors, not all categories add up to *N* = 288, *n*: subgroup sample size for each factor.

**Table 3 nutrients-11-01897-t003:** Food Safety Knowledge Scores by Section.

Food Safety Topic	Items (*n*)	Average (%)	Standard Deviation (%)
General Food Safety	11	70.74	15.53
Cross-Contamination (Separation)	8	83.03	15.62
Food Preparation	8	73.70	20.43
Food Storage (Chill)	8	69.53	17.47
Clean Up (Cleaning/Hygiene)	10	77.64	17.88
Overall Food Safety Knowledge	45	74.77	12.24

*n*: number of items per food safety section in the questionnaire.

**Table 4 nutrients-11-01897-t004:** Food safety knowledge across socioeconomic groups of cancer patients.

Socioeconomic Factor	Food Safety Knowledge Scores			
	Categories	Average (%)	Standard Deviation (%)	Significance (*P*)
Income ($)	<$1000 *	68.16	17.93	<0.005
	<$2000	72.54	9.36	
	<$3000	74.01	15.68	
	<$4000 *	78.72	7.98	
	≥$4000 *	76.62	9.92	
Food Assistance (Federal)	SNAP *	66.76	16.95	<0.001
	Other	71.11	7.86	
	None *	75.94	11.21	
Food Assistance (Private)	Receive	65.63	16.18	<0.001
	Don’t Receive	75.77	11.39	
Food Security	Secure	75.96	11.05	<0.001
	Insecure	69.74	15.50	

* denotes groups with statistically different food safety knowledge scores.

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
