# Peer review of "Assessment of Food Safety Knowledge and Behaviors of Cancer Patients Receiving Treatment"

_nutrients, 2019, doi:10.3390/nu11081897_

Round 1

Reviewer 1 Report

Thank you for the opportunity to review this interesting manuscript, which describes the assessment of food safety knowledge and behaviors of cancer patients receiving treatment. I agree that having an understanding of patients’ knowledge, attitudes, behaviors, and risk  perceptions is vital to the development of appropriate food safety education to prevent the adverse effects that foodborne diseases could impose on this population. However, I have provided some comments and suggestion to improve the quality of presentation of the results:

ABSTRACT: Line 24: 74.77%±12.24% - % after 74.77 should be deleted (74.77±12.24%)

Line 25: % after 69.53 should be deleted. Please correct the same in the whole manuscript

Line 24 and 25: remove “only”

KEYWORDS: Please add: food safety behaviors or food safety knowledge,  foodborne infections or foodborne disease

RESULTS: All statistically important differences should be accordingly provided  in the tables and figures with results, not only in the results' description.

Line 258: Please remove sentence: “While not statistically significant, education yielded P=0.052.”

Line 262: Please provide explanation: (average ± standard deviation)

Line 277: Please remove sentence: “It should be noted that, while the raw food security score was significant, the food security status determined by the score was not significant (P=0.073) at α=0.05.”

Table 3. Please remove all % in table near the numeric values. It is enough that (%) appeared in the column headers.

DISCUSSION: The discussion need to be enriched in more studies from last decade. I suggest the Authors to underline the special importance of a proper diet as a cancer chemoprevention factor. Furthermore, at the end the Authors should propose some public health intervention could reduce unhealthy, risk dietary habits among this population.

Author Response

ABSTRACT: Line 24: 74.77%±12.24% - % after 74.77 should be deleted (74.77±12.24%)

Corrected

Line 25: % after 69.53 should be deleted. Please correct the same in the whole manuscript

Corrected

Line 24 and 25: remove “only”

Corrected

KEYWORDS: Please add: food safety behaviors or food safety knowledge, foodborne infections or foodborne disease

Added

RESULTS: All statistically important differences should be accordingly provided in the tables and figures with results, not only in the results' description.

Table 4 added to provide information about important differences across socio-economic factors.

Line 258: Please remove sentence: “While not statistically significant, education yielded P=0.052.”

Removed

Line 262: Please provide explanation: (average ± standard deviation) 

Explanation (average ± standard deviation) added.

Line 277: Please remove sentence: “It should be noted that, while the raw food security score was significant, the food security status determined by the score was not significant (P=0.073) at α=0.05.”

Removed

Table 3. Please remove all % in table near the numeric values. It is enough that (%) appeared in the column headers.

Removed

DISCUSSION: The discussion needs to be enriched in more studies from last decade. I suggest the Authors to underline the special importance of a proper diet as a cancer chemoprevention factor. Furthermore, at the end the Authors should propose some public health intervention could reduce unhealthy, risky dietary habits among this population.

Thank you for your comment.  We made additions to the discussion section to reflect on importance of the diet in treatment outcomes. We also made a recommendation for a public health intervention. Ln 346-355.

Reviewer 2 Report

1) Foodborne illness is not well-described. What does these ilnesses includes? This should be describe more specific in the Introduction. 

2) Figure 1 is unclear, 1a, 1b and 1c are difficult to read.

3) The discussion section should summarise the results more clearly and compare these with other studies. 

4) Conclusions should summarise and conclude the results linking to the larger context.

Author Response

Foodborne illness is not well-described. What do these illnesses include? This should be described more specifically in the Introduction. 

Thank you for your comment.  We made additions to the text in the introduction section as suggested. Ln 54-57 was added clarifying that “Foodborne diseases occur due to consumption of foods contaminated with foodborne viruses, bacteria, or micro parasites. The most common foodborne pathogens include norovirus, non-typhoidal Salmonella spp., Campylobacter spp. , and others. Listeria monocytogenes causes some of the most severe infections with up to 30% death rate [1].

Figure 1 is unclear, 1a, 1b and 1c are difficult to read.

Thank you for your comment.  We have increased the font and reorganized Figure 1 to make it clear and readable.

The discussion section should summarize the results more clearly and compare these with other studies. 

We have re-worded the discussion to summarize the findings more clearly. The text is edited to clearly emphasize our study findings and contrast them with previously published research.

Conclusions should summarize and conclude the results linking to the larger context.

Reviewer 3 Report

The article deals a novel topic and will be of interest to Nutrient readers. However, there are several aspects that need to be reviewed and modified:

Introduction

·         Bibliographic citation number 2 is used in the document to refer to the risk of pathogenic infection in cancer patients (lines 35-37 and lines 286-287). However, this bibliographic citation, which is generally used, is very specific since it refers to the risk of infection by Listeria monocytogenes in patients with cancer-blood.  

Methodology

·         With regard to the measuring instrument, it should be said whether it has been previously validated. If yes, give the corresponding bibliographical citations. The time of collection of the information should be explained (i.e. last 12 months, last month, etc.).

·         In addition, the questions are scored is a bit rare, since there are dichotomous questions, likert scale and finally a summation is made of all of them. It should be explained why it is scored this way.

·         Lines 118 and 119 explain the justification for measuring some variables, but they are not methodology. My recommendation is to use this text in the introduction part.

Results

·         I understand that any research work may have missing values. However, in this case, I consider that there should not be missing values in such important variables as sex, age or type of cancer.

·         The tables use abbreviations that should be explained, perhaps in a footnote. (i.e. H.S.).

·         Review the results of the variable "Employment status" in Table 1 (401=).

·         Figure 1 is of poor quality and should also be labelled with a), b) and c), according to the legend. As a recommendation, the quality may improve if the figure is divided.

·         Other recommendations: Although the sample size is low, it would be interesting to evaluate the results by sex or stage of the tumor, because there could be differences.

Discussion

Throughout the document the authors speak of "risk".  However, the wording should be well revised since we are dealing with a fully descriptive article, so that strict conclusions should be avoided.

Bibliography

Review the format

Author Response

Bibliographic citation number 2 is used in the document to refer to the risk of pathogenic infection in cancer patients (lines 35-37 and lines 286-287). However, this bibliographic citation, which is generally used, is very specific since it refers to the risk of infection by Listeria monocytogenes in patients with cancer-blood.  

Thank you for your comment.  We added additional citations to include other examples of cancer related susceptibilities for foodborne infections, Ln 35-42

With regard to the measuring instrument, it should be said whether it has been previously validated. If yes, give the corresponding bibliographical citations. The time of collection of the information should be explained (i.e. last 12 months, last month, etc.).

All measuring instruments have been previously published or validated, and referenced in the text. Ln 131, 138-139, 141, 144, and 154

Time of collection is stated in the text (Ln 109-110)

In addition, the questions are scored is a bit rare, since there are dichotomous questions, likert scale and finally a summation is made of all of them. It should be explained why it is scored this way.

Thank you for this comment. Dichotomous data and Likert data was summarized and analyzed separately. We have added the clarification in data analysis section Ln 167-168

Lines 118 and 119 explain the justification for measuring some variables, but they are not methodology. My recommendation is to use this text in the introduction part.

Thank you for your comment.  Those lines have been relocated to the introduction, Ln 76-77.

Results

I understand that any research work may have missing values. However, in this case, I consider that there should not be missing values in such important variables as sex, age or type of cancer.

Thank you for the comment. In our sample, one participant was unable to or chose not respond to gender question. Another person did not want to or skipped over sex question. Multiple people refused to respond/skipped over the type of cancer questions. Although the sample size being at 288, unfortunately he patients were actually receiving their treatment and it is possible that they were strained by the survey which have may lead to more incompletely answered questions.

The tables use abbreviations that should be explained, perhaps in a footnote. (i.e. H.S.).

Thank you for your comment.  Footnote explanations of H.S., GED, SNAP, WIC, and d/w have been added to the tables.

Review the results of the variable "Employment status" in Table 1 (401=).

Corrected

Figure 1 is of poor quality and should also be labelled with a), b) and c), according to the legend. As a recommendation, the quality may improve if the figure is divided.

-

Thank you for the comment. We have corrected the labelling, reorganized the figures and increased the font to improve the quality of the figure.

Other recommendations: Although the sample size is low, it would be interesting to evaluate the results by sex or stage of the tumor, because there could be differences.

Thank you for your comment.  Food safety knowledge and food acquisition practices were both analyzed by sex and stage of cancer but the effect was not significant. 

Discussion

Throughout the document the authors speak of "risk".  However, the wording should be well revised since we are dealing with a fully descriptive article, so that strict conclusions should be avoided.

Thank you for making this distinction. In food safety field the understanding of risk is broader than quantitative risk. Especially when we talk about risk perceptions, etc.  (For instance Chen et al. 2010, Fein et al. 2011, and other)

Also, the increased risks of contracting infections among cancer patients is well documented and quantified (for instance Lund et al. 2002, FAO/WHO Risk assessment of Listeria monocytogenes; 2004)

Bibliography

Review the format

The bibliography format was imported from the reference management software and manually checked for formatting errors. All errors were corrected.

Reviewer 4 Report

This study was aiming to determine current food safety knowledge, behaviors, and attitudes among cancer patients seeking treatment, and to assess the effect of sociodemographic factors, including food insecurity, on food safety risks. This report was well written. The data were well organized. The findings in present study will benefit healthcare providers to better educate patients in the food safety practices necessary to decrease risk of foodborne infection, and to provide targeted food safety education to low food security patients.

Author Response

Thank you for your comments, we very much appreciate it. We too hope that this work will lead to better education and behavior change interventions for cancer patients.

Round 2

Reviewer 1 Report

The manuscript has ben improved. I have no more suggestions.

Author Response

Thank you very much for your feedback.

Reviewer 2 Report

Table 1: Number of patients aged 50 or over is 222, in the text:  most were over the age of 50 (77.6%, n=222) --> please correct another

Table 1: What does mean Employment status 40+ or ≤40 ? Additionally, please delete = after 401

Table 1: Monthly income units could be after the sentence "What is your household’s monthly income? ($)" 

Figure 1: Why does the scale is different in section a compared with b and c. Could it be disagree/agree in all sections 1,2 and 3?

Table 4: Income unit is missing. Should the categories be <1000, ≥1000 and <2000, 2000 and <3000, etc.

Author Response

Table 1: Number of patients aged 50 or over is 222, in the text:  most were over the age of 50 (77.6%, n=222) --> please correct another

Corrected to “most were aged 50 years or older (77.1%, n=222)” Ln 175-176

Table 1: What does mean Employment status 40+ or ≤40 ? Additionally, please delete = after 401

Employment status has been clarified and = has been corrected.

Table 1: Monthly income units could be after the sentence "What is your household’s monthly income? ($)" 

Corrected

Figure 1: Why does the scale is different in section a compared with b and c. Could it be disagree/agree in all sections 1,2 and 3?

Thank you for your comment.  The scale is different in the figure because the scale is also different in the questionnaire, since the survey is adapted from a collection of validated tools used in previous studies.

Table 4: Income unit is missing. Should the categories be <1000, ≥1000 and <2000, ≥2000 and <3000, etc.

Corrected

Reviewer 3 Report

Few of the variables analyzed contain the information of the 288 participants, so the missing values should be better explained.

Author Response

Thank you for your comment.  A clarification has been added to the methods section stating that “Although the sample size was 288, patients were actively receiving treatment at the time of survey administration.  Therefore, the survey may have presented strain to the patients, which may have led to incomplete surveys and questions with some missing data.” Ln 157-160.